# Study on the Antioxidant Effect of Shikonin-Loaded β-Cyclodextrin Forming Host–Guest Complexes That Prevent Skin from Photoaging

**DOI:** 10.3390/ijms242015177

**Published:** 2023-10-14

**Authors:** Yan Yue, Yuqing Fang, Ruoyang Jia, Keang Cao, Xue Chen, Hongmei Xia, Zhiqing Cheng

**Affiliations:** College of Pharmacy, Anhui University of Chinese Medicine, Hefei 230012, China; yueyan@stu.ahtcm.edu.cn (Y.Y.); 1312649763@ahtcm.edu.cn (Y.F.); 2022205221003@stu.ahtcm.edu.cn (R.J.); 2023205221003@stu.ahtcm.edu.cn (K.C.); 2023205221004@stu.ahtcm.edu.cn (X.C.); 1312649763@stu.ahtcm.edu.cn (Z.C.)

**Keywords:** shikonin-loaded β-cyclodextrin, antioxidant, skin photoaging, SOD activity

## Abstract

When the skin is overexposed to ultraviolet rays, free radicals will accumulate in the skin, causing lipid damage and even inducing photoaging of the skin. Combination therapy with antioxidant drugs is a good choice for topical treatment of skin photoaging due to its special physiological structure. In this paper, shikonin was encapsulated in β-cyclodextrin (SH-β-CD) by the precipitation crystallization method, which delayed the release of the drug and increased drug solubility. The average diameter of SH-β-CD was 203.0 ± 21.27 nm with a zeta potential of −14.4 ± 0.5 mV. The encapsulation efficiency (EE%) was 65.9 ± 7.13%. The results of the in vitro permeation across the dialysis membrane and ex vivo transdermal release rates were 52.98 ± 1.21% and 88.25 ± 3.26%, respectively. In vitro antioxidant and antilipid peroxidation model assay revealed the antioxidant potential of SH and SH-β-CD. In the mice model of skin photoaging, SH and SH-β-CD had a recovery effect on the skin damage of mice, which could significantly increase the superoxide dismutase (SOD) activity in the skin. Briefly, SH-β-CD had an obvious therapeutic effect on the skin photoaging of mice caused by UV, and it is promising in skin disease treatment and skin care.

## 1. Introduction

The damage caused by excessive exposure of the skin to ultraviolet (UV) light is called skin photoaging [1]. This is a complex process, in which many mechanisms play a role at the same time [2]. These mechanisms may include genome mutations, accumulation of toxic metabolites, hormone deficiency, increase in reactive oxygen species (ROS), and macromolecular cross-linking caused by glycosylation [3]. From a histological perspective, it is mainly manifested by the accumulation of elastin material below the epidermal dermal connection [4]. At the molecular level, UV radiation leads to a significant accumulation of ROS in vivo, which can activate cell surface receptors on keratin-forming cells and fibroblasts in the skin, initiate a signal transduction cascade, and induce the activation of several transcription factors including activator protein-1 (AP-1) [5]. AP-1 induces upregulation of matrix metalloproteinases (MMPs), which further leads to the breakdown of collagen in the extracellular matrix and the shutdown of new collagen synthesis [6]. This could lead to dry skin, decreased elasticity, sagging, and might increase the risk of melanoma [7]. As generally known, sunscreen has some protective effect on skin exposed to ultraviolet light. In addition, local administration of drugs such as glucocorticoids and immunosuppressants also has different degrees of adverse reactions, and not suitable for long-term use. Therefore, how to design a new drug delivery against skin photoaging is very necessary [8].

Shikonin (SH, C_16_H_16_O_5_, 288.31) is a purple powder mostly found in *Lithospermum erythrorhizon* Sieb. et Zucc. [9]. SH appears as a kind of needle-like crystal observed under a microscope [10]. A large number of clinical studies have shown that SH as a compound of the naphthoquinone class has a series of pharmacological activities such as antiproliferation [11], anti-inflammation [12], antiangiogenesis [13], and others [14]. Studies have shown that it could effectively inhibit the activation of nuclear factor kappa B (NF-κB) signal converter and transcription activator 3(STAT-3), so it has good application in the treatment of various skin diseases [15]. However, due to its poor water solubility and other limitations in transdermal delivery, it is necessary to create a suitable carrier to overcome these drawbacks [16].

According to the number of pyranose-glucose units, cyclodextrins (CDs) are a series of circular oligosaccharides formed by the α-1, 4-glycosidic bond of the D-glucopyranose, resembling a hollow cavity [17]. Compared with α-CD and γ-CD, β-CD has a moderate-size cavity, and is easy to form inclusion complexes, which have been widely studied [18,19]. In an aqueous solution, due to hydrogen bonding between β-CD molecules and water molecules, β-CD molecules trap the poor water-soluble guest molecule in the cavity in β-CD, forming a stable host–guest molecule [20]. The hydrophobic groups of the SH molecule can interact with the hydrophobic groups inside the cavity in β-CD to generate a complex and improve its solubility, stability, antioxidant activity, and bioavailability [21,22].

SH can protect the brain from ischemic injury by inhibiting MMP-9 overexpression by controlling inflammatory responses and enhancing the permeability of the blood–brain barrier (BBB) [23]. SH is also thought to inhibit the expression of MMP-2 and MMP-9, which may inhibit glioma cell migration and invasion by inhibiting P13K/Akt signaling. Therefore, SH can be used as an MMPs inhibitor to limit MMPs expression and reduce collagen breakdown in a photoaging environment, thereby suppressing skin redness, dryness, and wrinkles caused by photoaging.

In this study, SH-β-CD was prepared using precipitation recrystallization and loading the guest molecule SH into β-CD, which was used as the host molecule. It was expected that the properties of β-CD would improve the solubility and stability of SH, while the transdermal penetration, retention function, and inhibition of skin photoaging by SH-β-CD would be also fully improved.

## 2. Results and Discussion

### 2.1. Apparent Stability Constant of SH-β-CD Was Determined

From Figure 1, it was found that the concentration of SH showed a linear increasing trend with the increase in β-CD concentration. According to Higuchi and Connors’ theory, this phase solubility diagram can be attributed to the A_L_ type, indicating an inclusion compound with a ratio of host to guest of 1:1 and improved solubility in water. According to Equation (1), the stability constant of SH-β-CD is 421.39 Lmol^−1^. The larger the k value, the more stable the generated inclusion compound.

### 2.2. Characterization of SH-β-CD

#### 2.2.1. Observed under Cryo-Electron Microscopy (Cryo-EM)

The shape of SH-β-CD under cryo-electron microscopy appears to be spherical in structure and with a uniform size (Figure 2A). The β-CD cavity is hydrophobic and can encapsulate hydrophobic components. SH is strongly hydrophobic and easily encapsulated into the β-CD cavity (Figure 2B). A strong hydrogen bond is formed between the hydroxyl hydrogen of SH and the oxygen of the glucose unit in β-CD. It indicates that SH and β-CD can form stable inclusion complexes.

#### 2.2.2. Particle Size and Zeta Potential

The particle size of SH-β-CD was 203 ± 21.27 nm (Figure 2C). The zeta potential of SH-β-CD was negative −14.4 ± 0.5 mV (Figure 2D), and the inclusion complexes were shown to have the same charge and mutual repulsion, thus ensuring that they existed in a uniformly distributed form (Figure 2E). The particle size and zeta potential of the SH-β-CD suspension did not change much during 4 weeks in the refrigerator at 4 °C, indicating that the SH-β-CD is stable.

#### 2.2.3. Encapsulation Efficiency (EE)

Taking the concentration (C) as the abscissa and the absorbance (A) as the ordinate, the standard curve was obtained: A = 22.933 × C − 0.0424 (R = 0.9997). The results showed that there was a good linear relationship between SH concentration and absorbance in the range 6–30 μg/mL. The precision and stability of SH were good because RSD was less than 1%.

There are many methods to determine the EE of SH-β-CD. Among them, the dialysis method utilizes the difference in molecular size of semipermeable membranes to achieve separation by timely replacement of the external phase, but suffers from the disadvantages of time-consuming and easy drug leakage. Gel column chromatography adopts the difference in relative molecular mass for separation, but suffers from long elution time and low drug concentration. The advantage of ultracentrifugation is cost saving, and it is more suitable for separating fat-soluble substances such as SH. The encapsulation rate for SH-β-CD was measured by ultracentrifugation and release of SH from the encapsulated complex with ethanol. The results of UV detection showed that the EE% of SH-β-CD was 65.9 ± 7.13%. These data indicated that SH was successfully encapsulated in β-CD with high EE%.

### 2.3. Molecular Docking

During skin photoaging, UV radiation causes upregulation of MMPs [24], which further leads to the downregulation of collagen and skin aging [25]. Therefore, we conducted a docking model of MMP_S_ and SH (Figure 3). SH maintains the traditional hydrogen bond with the amino acid residue GLU1 of MMP_S_, and produces hydrophobic interaction with Val5, LYS3, and Leu371. In addition, it forms a major adverse collision with SER25.

### 2.4. Absorption, Distribution, Metabolic, Excretion, and Toxicity—(ADMET) Prediction

In this study, the SwissADME web tool was used to study the pharmacokinetic characteristics of SH and β-CD (Table 1). The potential of SH as a promising treatment for skin photoaging was identified by predicting all of the ADMET properties for SH. As can be seen from Table 1, the bioavailability of SH was relatively low and the absorption level of SH is level III.

### 2.5. In Vitro and Ex Vivo Penetration of SH and SH-β-CD

The drug release rate is an important characteristic in evaluating the drug delivery system. In addition, it has different release behavior in the penetration across the dialysis membrane and mouse skin.

#### 2.5.1. Penetration of SH and SH-β-CD across the Dialysis Membrane In Vitro

The in vitro release behavior of SH and SH-β-CD was quantitatively studied by the Franz diffusion method. Due to the dialysis membrane being able to intercept molecules of a certain size, the biomacromolecules in the sample solution are trapped on the dialysis membrane, while the small molecule drugs are constantly diffusing and entering the lower buffer solution. The drug is transported from the high-concentration side to the low-concentration side according to the concentration gradient. With the increase in diffusion time, SH in the upper diffusion pool gradually diffused into the PBS release medium in the receiving pool. At 96 h, the cumulative permeability of the SH group was 62.98 ± 1.21%, while SH-β-CD was 48.38 ± 3.09% (Figure 4). This may be because the large-molecular-weight β-CD was trapped outside the dialysis membrane, while the small-molecular-weight free SH penetrated more easily.

#### 2.5.2. Penetration of SH and SH-β-CD across the Mouse Skin Ex Vivo

Encapsulation of SH in β-CD improves drug stability and enables efficient loading. This can be verified in transdermal experiments. The penetration rate of SH and SH-β-CD across mouse dorsal skin was studied. When the drug diffused, it needed to cross multiple barriers in turn. The cumulative permeability of the SH group and SH-β-CD was 38.25 ± 3.26% and 42.27 ± 0.96% at 96 h (Figure 4). The results showed that β-CD could wrap SH and stay in the skin, and then release slowly, which made it possible to use it as a transdermal drug delivery system. Because β-CD is a macromolecule, after encapsulation of SH, the SH-β-CD is also a macromolecule. Since the macromolecule passes through the cell membrane by way of cytophagy, this shortens the time required for transmembrane transport and allows SH to be further released from the encapsulated compound, leading to the osmotic effect of SH-β-CD.

### 2.6. Pharmacokinetic Studies

To determine the release mechanism of SH and SH-β-CD, Origin 2016 software was used to simulate the release curve, and common models such as zero-order kinetics, first-order kinetics, Higuchi, Weibull CDF and Hixson–Crowell, were used to analyze the drug release kinetics. Among them, Table 2 lists the correlation coefficient (R) value. The Weibull CDF model has the highest r-value, suggesting that it is more suitable for this model and has a linear relationship.

### 2.7. Antioxidant Assay

Many diseases are related to the oxidative damage caused by free radicals. When the organism is damaged, in pathological conditions, oxidative stress damage and other reactions occur, free radicals are generated in the cells, which in turn have further toxic effect on the cells. SH as a natural antioxidant can scavenge free radicals and protect cells from oxidative damage. The scavenging H_2_O_2_ and DPPH (1,1-diphenyl-2-picrylhydrazyl) assays are the most effective, convenient, and accurate methods for evaluating the antioxidant activity of chemical compounds.

#### 2.7.1. Scavenging Rate of SH and SH-β-CD on ·OH Free Radicals of H_2_O_2_

Various reactive oxygen species are metabolized in the body, including hydrogen peroxide. Hydrogen peroxide is produced when electrons are transferred to oxygen in the respiratory chain of the body. Under normal conditions, the enzyme catalase is present in the body, which rapidly breaks down hydrogen peroxide and prevents it from causing damage to cells. However, when the body is in pathological conditions, hydrogen peroxide is difficult to break down and remove, thus producing toxic effects on the cells, such as damage to the endothelial cells, resulting in cardiovascular and cerebrovascular diseases, etc.

At the same concentration of H_2_O_2_ (1.10 mg/mL), the scavenging ability follows the order: SH group > SH-β-CD group > B-β-CD group (Figure 5A). Additionally, using different SH concentrations (C) to perform linear regression on the scavenging rate (E%) on H_2_O_2_, the regression equation was obtained: E = 24.74 × C − 0.9024 (R = 0.9992). The experiment showed that SH concentration was linearly correlated with scavenging H_2_O_2_ rate in the range 0.22–2.2 mg/mL (Figure 5B). According to the regression equation, IC_50_ of the scavenging rate of SH on H_2_O_2_ was 2.05 mg/mL.

#### 2.7.2. Scavenging Rate of SH and SH-β-CD on Free Radicals of DPPH

The scavenging DPPH radical assay is commonly used to assess antioxidants. Antioxidants have the hydrogen donor ability to reduce the free radical DPPH to a stable form and reduce the absorbance of the DPPH solution at 517 nm. The scavenging ability for the same concentration on DPPH: SH group > SH-β-CD group > B-β-CD group (Figure 5C), indicating that SH had a stronger scavenging rate (*p* < 0.001), and SH-β-CD might be due to the cavity structure of β-CD, which means the SH cannot be completely released in the same time, thus making its scavenging rate on DPPH lower. E% of SH solution at different concentrations (0.176–2.20 mg/mL) increased with the increase in SH concentration, as shown in Figure 5D.

#### 2.7.3. Inhibitory Effect of SH-β-CD on MDA in Mouse Organs

Lipid peroxidation refers to the oxidative deterioration of polyunsaturated fatty acids and lipids, which may lead to changes in cell membrane fluidity and permeability, damage to DNA and protein, and then affect the normal physiological functions of cells [26]. Studies have shown that many human diseases, such as tumors, arteriosclerosis, and aging, are related to lipid peroxidation. Among them, malondialdehyde (MDA) is one of the most important products of membrane lipid peroxidation [27]. There are two ways of production (Figure 6), one is to degrade arachidonic acid (AA) through enzymatic reaction and the other is produced by nonenzymatic oxidative degradation of polyunsaturated fatty acids [28,29].

Under the condition of being irradiated by long-term UV, the oxygen free radicals produced by the organism attack polyunsaturated fatty acids (PUFA) in biofilm, and then a large amount of lipid hydroperoxide is generated through lipid peroxidation, which changes the skin color, thickness, and texture. At the same time, lipid peroxidation also forms toxic substances such as MDA, which leads to further harmful skin problems and may even induce skin cancer [30]. As one of the final products of oxidation, MDA can be used to judge the degree of damage caused by free radicals [31]. Therefore, the determination of MDA is widely used for the level of lipid oxidation or peroxidation. The reaction principle is that MDA can react with TBA in high-temperature and acidic environments to form a stable red MDA-TBA compound. In addition, UV may also disrupt the telomere loop, expose the TTAGGG overhang, and promote aging. Intrinsic aging is complemented by repeated cell divisions that shorten telomeres [32]. Overall, increased levels of ROS may modulate signal transduction cascades, including involvement in tumor suppressor gene p53 and mitogen-activated protein kinase (MAPK) pathways, and lead to upregulation of AP-1 and NF-κB, and downregulation of transforming growth factor (TGF-β) [33,34]. NF-κB regulation increased the levels of interleukin-1 family (IL-1) and tumor necrosis factor (TNF-α), while AP-1 upregulated the activation of MMPs [35].

During the experiment of lipid peroxidation in isolated mouse tissues, Fe^2+^ was added as induction. The results showed that when five different concentrations of SH solutions (0.22, 0.55, 1.10, 1.65, and 2.2 mg/mL) reacted with liver, kidney, and brain homogenate (Figure 7A), the inhibition rate was dose-dependent on the whole, and the ability of inhibited MDA production was liver > kidney > brain, which indicated that the action of SH was somewhat organization targeted. In addition, in each group of homogenates, the inhibition rate for B-β-CD group < SH-β-CD group < SH group (*p* < 0.001) (Figure 7B–D), which indicated that the antioxidant effect of SH-β-CD was lower than that of SH, which was consistent with the experimental results of anti-DPPH and anti-H_2_O_2_. It may be that SH was encapsulated in the cavity structure of β-CD, which indicates that SH cannot release completely in a short time, and played a role of slow release and controlled release, thus inhibiting lipid peroxidation.

### 2.8. Pharmacodynamic Studies of Skin Photoaging

The establishment of photoaging the mouse model and the determination of administration time are shown in Figure 8. The macroscopic improvement effect of the SH, B-β-CD, and SH-β-CD groups on the skin photoaging in mice can be seen in Figure 9A. The representative pictures listed show conditions at the end of the experiment. The mice in the blank group, which only underwent skin depilation on the back, indicated the fine texture of normal skin in the state of smooth, red, lustrous, and no sagging phenomenon. In contrast, it was found that in the model group, mice began to show wrinkles on the skin, visible to the naked eye in the fourth week, and the skin became coarse and reddish until the eighth week. When the skin already showed the typical photoaging phenomenon, with thickened and deepened wrinkles, obviously becoming rough, the phenomenon of skin damage appeared in serious cases. From the picture of skin damage, except for the model group, the skin damage in mice in the B-β-CD treatment group was the most serious, followed by the SH treatment group and the SH-β-CD treatment group, which were the less severe. Additionally, the transdermal penetration rate was greater in the SH-β-CD group than in the SH group, which confirms this. Furthermore, reduced melanin synthesis can be supported by a variety of mechanisms, such as blocking tyrosinase transcription, dispersion of keratinocyte cytochrome particles, or enhanced turnover of epidermal cells.

The macroscopic scoring results for each group showed (Figure 9B) that the model group was already significantly different from the blank group from the fourth week, indicating successful modeling, and the scores for each dosing group were statistically significant. The skin scores for mice in the model group were significantly higher than those in the blank group, indicating an increase in the severity of skin photoaging. After 8 weeks of treatment with SH, B-β-CD, and SH-β-CD, the scores did decrease significantly, suggesting that the severity of inflammation was effectively alleviated.

As shown in Figure 9C, the skin thickness in mice in the blank group was 0.85 ± 0.5 mm and the model group had the thickest skin with a rough surface and increased folds. After administration, the skin thickness in the SH and SH-β-CD treatment groups was smaller than that in the model group, while the thickness in the SH-β-CD group was thinner than that in the SH group (*p* < 0.01). The inflammatory symptoms of mice were significantly alleviated, indicating that SH and SH-β-CD had better therapeutic effects on UV-induced skin photoaging in mice.

In addition, the weight of the mouse spleen could also be used as an indicator of the severity of inflammation [27]. The changes in spleen index in photoaged mice are shown in Figure 9D. The proportion of spleen in the model group was the largest. It suggests splenomegaly caused by inflammatory symptoms of photoaging. According to the experimental results, the degree of splenomegaly follows: blank group < SH-β-CD group < SH group < β-CD group < model group (*p* < 0.001).

SH-β-CD had a better inhibitory effect on serum low-density lipoprotein (LDL) than the SH group at the same concentration, which indicated the clearance rate for SH-β-CD on LDL was improved after SH was coated with β-CD (Figure 9E). At the same time, SH could promote the expression of LDL receptors and reduce the level of serum LDL, further reducing inflammation. In addition, normal LDL is less likely to cause diseases such as atherosclerosis [36,37]. However, high levels of endogenous LDL are easily modified by oxidation to form oxidized LDL, which is associated with hypercholesterolemia [38]. Additionally, oxidized low-density lipoprotein has a high degree of immunogenicity, stimulating the body to produce some autoantibodies, Oxidized low-density lipoprotein (Ox-LDL) has been found in a variety of lesions [39]. It has been reported that in human acute myelogenous leukemia (AML) cells, the metabolic rate for LDL was significantly enhanced. Experiments have shown that effective lowering of serum LDL levels was very important to treat the disease.

SOD (superoxide dismutase) is an important part of the first-line defense system against reactive oxygen species [40]. It is an endogenous antioxidant enzyme with O^2-^ as substrate, and O^2−^ and H^+^ are converted into H_2_O_2_ and O^2-^ under its catalytic action, thus reducing the damage of O^2−^ as potentially harmful to the body and ending the chain reaction of oxidation of reactive oxygen species in the body [41]. The activity of SOD in the control group and the drug group was higher than that in the model group, and the activity of SOD in the control group was the highest (Figure 9F). It could be speculated that excessive UV promoted the production of a large amount of active oxygen in mice, and the increase in active oxygen concentration also induced the increase in SOD activity [42]. In this study, the activity of endogenous SOD in mice was inhibited to varying degrees due to the oxidative stress induced by UV, which might be because the skin was also the first line of defense against the invasion of harmful substances in mice, and the antioxidant system in tissues was seriously damaged [43]. At the same time, oxidative stress promoted the synthesis of antioxidant enzymes, induced the increase in antioxidant enzyme activity, and effectively reduced the biofilm damage caused by lipid peroxidation [44].

These observations confirmed the superiority of SH-β-CD and its favorable aspects when applied topically before UV irradiation. The better antiwrinkle effect of the designed SH-β-CD might be due to the successful transfer of SH into the dermis, while the hydrophobic nature of SH might further enable its deposition into the upper dermis, where it exerted successive topical drug effects to combat more effectively the loss of skin elasticity and collagen loss.

The results show that SH-β-CD had a therapeutic effect on mice exposed to UV rays for a long time, and it could effectively play a better antioxidant role and prevent skin from pathological damage. Therefore, it is feasible to prepare SH into SH-β-CD for the treatment of skin photoaging.

### 2.9. Hematoxylineosin Staining Study (H&E) Study

Morphological changes in skin tissues were observed by the H&E staining method and histological evaluation was performed. As shown in Figure 10, compared to the normal group, the model group showed extensive abnormal proliferation of keratinocyte cells, leading to severe keratinization. In addition, a large number of capillary hyperplasia were found, indicating that UV led to severe infiltration of inflammatory cells and skin damage [45]. The SH and SH-β-CD treatment greatly inhibited the hyperproliferation and inflammatory infiltration of keratin-forming cells, and after SH treatment, the inflammatory response was not significantly reduced and severe tissue damage was still visible, whereas the SH-β-CD group possessed minimal pathological features and did not differ significantly from the normal group in terms of skin status. This result suggests that SH was enveloped in the β-CD and formed SH-β-CD, which could increase the surface area of the encapsulated drug and be more conducive to better drug apposition in the surface folds of skin in the photoaged mice. This allowed better surface attachment of the encapsulated drug to the stratum corneum and keratinocyte clusters, which might lead to several hours of accumulation, resulting in sustained drug release. Thus, SH-β-CD could be skin-targeted.

## 3. Materials and Methods

### 3.1. Materials and Animals

SH (purity 98.0%) was purchased from Chengdu Health Biotechnology Co., Ltd. (Chengdu, China). Absolute ethanol was purchased from Fujie Chemical Reagent Company (Shanghai, China). β-CD was purchased from Yuanfeng Biology Science and Technology Company (Shanghai, China). 1,1-diphenyl-2-picrylhydrazyl (DPPH) was purchased from Runjie Chemical Reagent Co., Ltd. (Shanghai, China). Hydrogen peroxide (H_2_O_2_) was provided by Shanghai SuYi Chemical Reagent Co., Ltd. (Shanghai, China). FeSO_4_∙7H_2_O and Thiobarbituric acid (TBA) were provided by Shanghai Yuanye Biology Science and Technology Co., Ltd. (Shanghai, China). Trichloroacetic acid (TCA) was provided by DAMAO Chemical Reagent Factory (Tianjin, China). The superoxide dismutase (SOD) kit was purchased from Jiancheng Biology Engineering Institute (Nanjing, China). Ethyl carbamate was purchased from Shanghai Suyi Chemical Reagent Co., Ltd. (China, Shanghai). All chemical reagents were of analytical grade.

Healthy female Kunming mice (20 ± 2 g) were purchased from the Animal Experimental Center of Anhui University of Chinese Medicine (Hefei, China). Animals were housed at a controlled temperature of 20–22 °C, relative humidity of 50–60%, and 12 h light–dark cycles. All animal experiments were conducted under the guidelines approved by the ethics committee of Anhui University of Chinese Medicine (Hefei, China).

### 3.2. Preparation of the Shikonin-Loaded β-Cyclodextrin of (SH-β-CD)

B-β-CD was prepared using a precipitation crystallization method. Firstly, 1.20 g of β-CD was weighed and dispersed evenly in 10 mL PBS (pH = 7.4). Then, 0.55 g of SH was weighed and fully dissolved into the absolute ethanol to obtain the SH solution. After that, 1 mL SH solution was added to -β-CD, stirred at 60 °C at 500 rpm/min for 2 h, and refrigerated at 4 °C for 24 h to obtain SH-β-CD with a concentration of 1.10 mg/mL. In the same way, blank β-cyclodextrin (B-β-CD) was prepared without SH.

### 3.3. Apparent Stability Constant of SH-β-CD Was Determined

A phase solubility diagram was used to determine the stability constant. Firstly, excess SH (0.5 mL) was added to 10 mL of β-CD solution containing different concentrations of β-CD, where the concentration of β-CD was 0, 10, 20, 30, 40, and 50 mmol·L^−1^. Afterward, it was fully shaken at room temperature and reached equilibrium, the lower aqueous solution was diluted, and SH content was measured at 516 nm. The experiment was repeated three times. According to Equation (1), the stability constant *Kc* was calculated from the slope and intercept of the linear part of the phase solubility diagram.
(1)$$Kc=\frac{K}{S_{0}\left(1-K\right)}$$

*Kc* is the stability constant of the inclusion compound at room temperature, and *S*_0_ is the solubility of SH in water without β-CD.

### 3.4. Characterization of the SH-β-CD 

#### 3.4.1. Observed under Cryo-Electron Microscopy (Cryo-EM)

Cryo-electron microscopy can reveal the diversity of particle size and shape. The morphology of SH-β-CD was examined under Cryo-EM (Glacios-200 KV, Thermo Fisher Scientific, 168 Third Avenue, Waltham, MA, USA).

#### 3.4.2. Particle Size Measurement and Zeta Potential 

The size and zeta potential of the inclusion compound are the key parameters to indicate its physical stability. A proper amount of SH-β-CD was diluted in PBS, and the size and zeta potential (*n* = 3) of the inclusion compound was measured by a particle size analyzer (Malvern Instruments Ltd., Malvern, UK).

#### 3.4.3. Encapsulation Efficiency (EE)

To a 10 mL volumetric flask, 1 mL of SH-β-CD solution was added, the volume was fixed with PBS solution, and shaken well. Next, 3 mL of sample was centrifuged at 4000 rpm for 10 min, after which the supernatant absorbance at 516 nm was measured. The data were brought into the standard curve to calculate the concentration of free drug (*C*_0_). In the next step, 1 mL of SH-β-CD solution was added to ethanol solution to fix the volume at 25 mL, the cyclodextrin shell was destroyed by ultrasonic sonication for 30 min; an appropriate amount of this solution was centrifuged at 4000 r/min for 10 min. The 516 nm absorbance of the supernatant was measured and the total drug concentration (*C_t_*) calculated by the standard curve: A = 22.933 × C − 0.0424 (R = 0.9974, *n* = 6). The drug encapsulation rate was calculated according to Equation (2):(2)$$EE\%=\frac{\left[C_{t}-C_{0}\right]}{C_{t}}\times 100\%$$
where *C_t_* is the total concentration of SH and *C*_0_ is the concentration of free SH.

#### 3.4.4. Stability of SH-β-CD

SH-β-CD was refrigerated at 4 °C and stored aseptically to determine the stability of SH-β-CD. The samples in the appropriate state of storage were taken at 0, 1, 2, 3, and 4 weeks to determine the particle size of the SH-β-CD suspension.

### 3.5. Molecular Docking 

Molecular docking is a method of drug design based on the characteristics of the receptor and the way the receptor interacts with drug molecules [46]. A theoretical model characterizes molecular interactions (ligands and receptors) and predicts binding patterns and affinity [47]. In the present study, molecular docking was performed using the Discovery Studio 2016 software [48]. The structure of MMP_S_ was derived from the protein database (PDB ID = 1 JZH). A docking study was conducted between SH and MMP_S_. Computer software (Discovery Studio 2016) was used to analyze the interactions between the individual molecules. The details of how Discovery Studio 2016 works follow:

Obtain the receptor from the PDB database;

(1)The protein is then prepared for preprocessing using the prepared protein function in the macromolecule module, which organizes the imported protein structure for further modeling operations;(2)The small molecule SH is prepared, and then Prepare Ligands in the small molecule module is opened to optimize the ligands. This function prepares ligands for the input of other protocols.(3)The binding site is then defined.

### 3.6. ADMET Prediction

ADMET profiles were analyzed using ADMET structure–activity relationship (admetSAR) 2.0 tool/database (http://lmmd.ecust.edu.cn/admetsar2/, accessed on 11 October 2023) access of 18 August 2023 [49] and an online version of SwissADME web tool (http://www.swissadme.ch, accessed on 11 October 2023) [50]. For analysis, the input of ligand molecules was formatted from the PubChem database. The level of lipophilicity was analyzed based on the atomic logarithm of (AlogP) [51]. The drug distribution for the blood–brain barrier (BBB) was examined. The cytochrome P450 (CYP) model (CYP1A2, CYP2C19, CYP2C9, CYP2D6, and CYP3A4) was used to evaluate substrates or inhibit drug metabolism. In addition, we also analyzed drug toxicity, mainly considering AMES toxicity and hepatotoxicity. In summary, all significant ADMET parameters of the compounds were estimated and examined.

### 3.7. Drug Release under Different Conditions

#### 3.7.1. In Vitro Drug Release Studies

The in vitro release pattern of SH and SH-β-CD was performed using Franz diffusion pools. The lower pool in each diffusion bottle was filled with PBS containing 10% ethanol as diffusion medium, and the MD34 dialysis membrane (MWCO: 8000–14,000) was placed above the lower pool, the temperature set at 37 ± 0.5 °C, and stirred at 300 rpm/min. The experiments were divided into the SH, SH-β-CD, PBS, and B-β-CD groups.

Then, 2.0 mL of sample was added into the upper pool and 2.0 mL of sample taken from a lower pool at 5, 10, 20, 30 min and 1–12, 24, 48, 60, 72, 84, and 96 h; 2.0 mL of PBS was injected into the lower pool after every sampling was completed. The absorbance was measured at 516 nm and recorded (*n* = 3). The amount of SH released at each time point was calculated using Equation (3):(3)$$Q(\%)=\frac{C_{n}\times V+\sum_{i=1}^{n-1}C_{i}\times V}{Q_{t}}$$
where *Q* is the cumulative drug release rate of SH at the different sampling points, *C_n_* is the mass concentration of SH at the nth time point (mg/mL), *V_n_* is the volume of the solution sampled at the nth time point (mL), *C_i_* is the mass concentration of SH in the receiving solution at the sampling point (*i* ≤ *n* − 1) (mg/mL), *V* is the total volume of the acceptor chamber (mL), and *Q_t_* is the theoretical drug content (1.10 mg/mL).

#### 3.7.2. Ex Vivo Drug Percutaneous Penetration Studies

To evaluate the accelerating effect of SH-β-CD on the percutaneous penetration of SH, drug release studies ex vivo were studied. The mice (20 ± 2 g) were raised adaptively for 3 days, and the back hair removed after 20% ethyl carbamate solution was injected into the abdominal cavity for anesthesia. After peeling off the skin, it was cleaned with PBS remove the remaining fatty tissue, and used immediately. The operation was similar to that described in Section 3.7.1 except that the dialysis membrane was replaced by the skin of mice (the epidermis layer was upward and the dermis layer was downward), and the temperature was 32 ± 0.5 °C.

#### 3.7.3. Release Kinetic Model of SH-β-CD 

The drug release kinetics and mechanism of SH and SH-β-CD were fitted by various mathematical models, including zero-order, first-order, Higuchi, Hixson–Crowel, and Weibull CDF [52]. The model with the highest correlation coefficient (R) was considered to be the best fit for the kinetic release. The equations of various dynamic models are provided in Table 3.

### 3.8. Antioxidant Activities Analysis

#### 3.8.1. Scavenging Effect on H_2_O_2_


The hydrogen peroxide solution was prepared by dissolving 0.1 mL hydrogen peroxide in 50 mL PBS. The experiment was divided into three groups: blank (*A*_0_), control (*A_c_*), and sample (*A_s_*). The *A*_0_ group was a mixture of 0.6 mL PBS and 1.8 mL H_2_O_2_ solution, the *As* the group was a mixture of 0.6 mL samples (the concentration of SH was 0.22, 0.55, 1.10, 1.65, and 2.20 mg/mL, concentration of B-β-CD was 1.10 mg/mL, and concentration of SH-β-CD was 1.10 mg/mL) and 1.8 mL PBS. In the A_c_ group, PBS was replaced with H_2_O_2_ solution. After the solutions of each group were reacted at room temperature for 10 min, the absorbance was measured at 230 nm, and the clearance rate on H_2_O_2_ by each sample was calculated using Equation (4).
(4)$$E\;\left(\%\right)=1-\frac{A_{s}-A_{c}}{A_{0}}\times 100\%$$
where *A_s_* is the absorbance of the sample solution, *A*_0_ is the absorbance of the blank control solution, and *A_c_* is the absorbance of the sample solution without H_2_O_2_.

#### 3.8.2. Scavenging Effects on DPPH Radicals

The activity of scavenging DPPH free radicals was measured according to the method reported in [53]; 8 mg of DPPH was weighed and 50 mL of absolute ethanol solution was used to fully dissolve, obtaining a 16% DPPH solution. The experiment was also divided into three groups: blank (*A*_0_), sample (*A_s_*), and control (*A_c_*). *A*_0_ was a mixture of 2.0 mL PBS and 1.0 mL DPPH solution, the A_s_ group was a mixture of 2.0 mL samples and 1.0 mL DPPH solution, and the A_c_ group was a mixture of 2.0 mL samples and 1.0 mL PBS. After mixing evenly, each group of solutions was placed in the dark to react for 30 min, and then the absorbance of the solutions at the absorption wavelength of 517 nm was measured. The scavenging activity on DPPH free radical was calculated according to Equation (4).

#### 3.8.3. Inhibitory Effect of SH-β-CD on MDA in Mouse Organs

Pretreatment of tissue homogenate: mice were fasted for 12 h, anesthetized with ethyl carbamate, and killed. The liver, kidney, and brain tissues were taken out quickly, rinsed repeatedly with cold normal saline until there was no blood stain, weighed and added to 9 times normal saline, crushed with a homogenizer, centrifuged at the speed of 4000 r/min for 15 min, and the supernatant taken to obtain 10% homogenate of each organ tissue. Samples were stored in a refrigerator at 4 °C for later use.

First, 1.0 mL of 10% tissue homogenate was added to the test tubes and divided into blank, model, and sample groups. Then, 0.1 mL of each sample (SH with the concentrations of 0.22, 0.55, 1.10, 1.65, and 2.20 mg/mL, SH-β-CD with the concentration of 1.10 mg/mL and corresponding B-β-CD) was added to the sample group, and added the same volume of physiological saline into the blank group and the model group. These were mixed evenly, and allowed to stand for 5 min. Next, 0.1 mL of 2.78% FeSO_4_ was added to the model group and the sample group, and 0.1 mL of normal saline was added to the blank group. After being mixed evenly, all groups were incubated at a constant temperature of 37 °C for 1.5 h. Then, 2.0 mL of 5.6% TCA was mixed well and placed in a 95 °C water bath for 40 min. Lastly, the samples were immediately cooled with running water, centrifuged at the speed of 4000 rpm/min for 8 min, the supernatant sucked and the absorbance measured at 532 nm, and the inhibition rate calculated on MDA production of each sample according to Equation (5):(5)$$The\;inhibition\;rate\;(IR,\%)=\frac{A_{c}-A_{s}}{A_{c}-A_{0}}\times 100\%$$
where *A*_0_ is the absorbance of the blank control group, *A_C_* is the absorbance of the model group, and *A_S_* is the absorbance of the sample group.

### 3.9. Pharmacodynamic Study of SH-β-CD on UV-Induced Skin Photoaging of the Mice

#### 3.9.1. Experimental Arrangement 

Ultraviolet rays (UV) are divided into three wavelengths: long-wave UVA (320–400 nm), medium-wave UVB (280–320 nm), and short-wave UVC (200–280 nm) [54]. It is found that UVA has strong penetrating power and can directly reach the dermis to destroy elastic and collagen fibers [55]. Additionally, most UVB can be absorbed by the epidermis, which is the main cause of epidermal cell decay [56]. Considering the effect of wavelength range on the skin photoaging model, a device constructed for this experiment used irradiation with two UVB lamps (45.1 cm, 580 uw/cm^2^) and one UVA lamp (30.2 cm, 580 uw/cm^2^) (Hong yuan Lighting Technology Co., Ltd., Wuhan, China). 

One week after adaptation to the environment, mice were randomly divided into 5 groups: blank, model, SH, B-β-CD, and SH-β-CD, with 10 mice in each group (Table 4). The mice were depilated on their backs, exposed to a 2 cm × 2 cm area of skin, and irradiated by UVA and UVB lamps except the control group. Mice received 30 min of radiation per day for the first week, followed by an additional 10 min of radiation per week, with the length of the stimulation period depending on the severity of the photoaging symptoms, ultimately set at 8 weeks. Starting from the first day of UV induction, mice in the SH, B-β-CD, and SH-β-CD groups were given 0.3 mL of samples, and mice in the control and model groups were treated with the same amount of PBS. The skin condition and skin thickness of mice were recorded and scored before daily administration (Table 5). On the second day after the last administration, the mice were anesthetized and killed. The blood, back skin, liver, and spleen were collected for later use.

#### 3.9.2. Biochemical Parameters Related to Skin Photoaging Resistance

##### Measurement of Spleen Index

The fresh spleen was taken from mice, rinsed repeatedly with normal saline until there was no blood on the surface, dried with filter paper, and weighed. The spleen index was calculated according to Equation (6):(6)$$spleen\;index=\frac{M_{p}}{M_{c}}\times 100\%$$
where *M_p_* is the mass of the spleen and *M_c_* is the mass of the mouse.

###### Determination of LDL Level in the Serum

Serum pretreatment: Blood was extracted from the mouse and the serum was obtained by centrifuging for 10 min at the speed of 1000 r/min after natural coagulation; 1 mL heparin citrate buffer (pH 5.04) per 100 µL of serum was added, mixed evenly, left at room temperature for 10 min, centrifuged at the speed of 1000 r/min for 10 min, adjusted the final pH to 5.1, and the precipitation weighed. The precipitation was suspended with a high-salt phosphoric acid buffer (pH 7.4) and twice the volume of the serum to dissolve the LDL precipitation. After 24 h of dialysis at 4 °C, the LDL extract was obtained and diluted with PBS containing 25 µg /mL LDL for later use.

The experiment was carried out according to the following steps:(1)The test tubes were divided into a blank, model, and sample groups by adding 1 mL of LDL extract solution;(2)1 mL of each sample (0.1, 0.2, 0.3, 0.4, and 0.5 mg/mL of SH, 0.5 mg/mL of SH-β-CD and corresponding B-β-CD) was added to the sample group, and 1 mL of PBS was added to the blank group and the model group;(3)0.2 mL FeSO_4_·7H_2_O (0.55 mmol/L) was added into each test tube of a model group and sample group, and the blank group was replaced with the same amount of PBS, mixed, and incubated at 37 °C for 3 h;(4)0.1 mL EDTA-Na_2_ (1 mmol/L) was added to terminate the reaction;(5)0.3 mL of each reaction suspension was taken and added to 2.5 mL TCA solution (20 g/dL) and 1 mL TBA solution (0.67 g/dL), and mixed in boiled water for 30 min;(6)The samples were cooled with running water, centrifuged at 3000 rpm/min for 10 min, and the supernatant was absorbed. The absorbance was measured at 550 nm. The inhibition rate for each sample on LDL was calculated according to Equation (4).

####### Determination of SOD Level in Skin

Ten percent skin homogenate was obtained according to that described in Section 3.8.3. The SOD activity of skin samples from the different groups was measured using the assay kits. The assay components were the blank, model, and drug groups, and 1 mL of reagent 1 application solution (0.1 mL of reservoir solution was taken and diluted to 1 mL with distilled water), 0.05 mL of skin homogenate supernatant, 0.1 mL of reagent 2, 0.1 mL of reagent 3, and 0.1 mL of reagent 4 application solution (prepared as reservoir solution: dilution solution = 1:14) were added to the reagent tubes of each group, and the reagent tubes of the control group were added with 1 mL of reagent 1 application solution, 0.05 mL of distilled water, 0.1 mL of reagent 2, 0.1 mL of reagent 3, and 0.1 mL of reagent 4 application solution to the control group. These were mixed thoroughly with a vortex mixer and placed in a constant temperature water bath at 37 °C for 40 min. Next, 2 mL of a color developer was added to each reagent tube (take 1 powder, add 75 mL of distilled water, heat to 70–80 degrees C to dissolve to make reagent 5, take 1 powder, add 75 mL of distilled water to make reagent 6, prepare with reagent 5: reagent 6: glacial acetic acid = 3:3:2), mixed well, left at room temperature for 10 min, and the absorbance at wavelength 550 nm was measured. SOD viability values (U) were calculated according to Equation (7).
(7)$$SOD\left(U/mgprot\right)=\frac{A_{0}-A_{S}}{A_{0}}/0.5\times \frac{V_{t}}{V_{s}}/P$$
where *A*_0_ is the control absorbance; *A_s_* is the measurement absorbance; *V_t_* is the total volume of the reaction system, 1 mL; *V_s_* is the volume of the sample added to the reaction system, 0.09 mL; and P is the protein content at the same homogenate concentration.

#### 3.9.3. H&E Study

The skins treated were cut into slices (50 μm) and fixed in 4% paraformaldehyde solution at 25 °C for 1 h, and then were placed in a refrigerator at 4 °C for 24 h. The sample was stained according to the microscopic observation and carried out after H&E staining. An illustration of H&E staining for skin lesions in each group was shown.

### 3.10. Statistical Analysis

All values are expressed as mean ± SD. Differences in the treatments are expressed as means and compared statistically using least significant differences (LSD) at the 5% level using the one-way ANOVA test with SPSS 26.0 software.

## 4. Conclusions

In this study, SH-β-CD was successfully prepared by a precipitation crystallization method, which could effectively encapsulate SH into β-CD to form the host–guest complexes and overcome the problem of low solubility of SH. In vitro free radical scavenging experiments showed that SH-β-CD has good antioxidant activity, and in vitro release studies showed that the drug release of SH-β-CD had a slow release property. In animal models of skin photoaging, it showed that SH-β-CD was better than the SH group for the inhibition of keratin formation and inflammatory infiltration. Furthermore, there was a significant inhibitory effect on MDA in vivo. In addition, SH-β-CD could enhance the activity of SOD and showed good therapeutic effect in UV-induced skin photoaging. In conclusion, SH could be retained in the skin or infiltrated directly into the systemic circulation for systemic treatment via dermal administration, which has great potential in the treatment of skin photoaging.

## Figures and Tables

**Figure 1 ijms-24-15177-f001:**
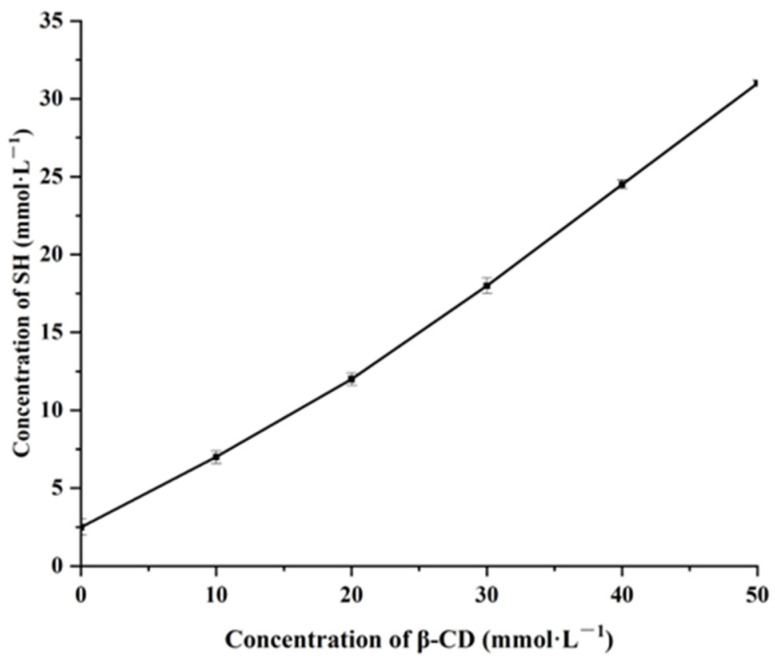
The apparent stability constant of SH-β-CD.

**Figure 2 ijms-24-15177-f002:**
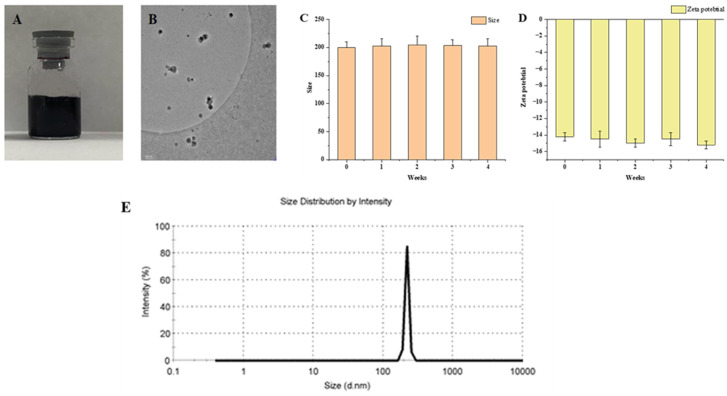
The characterization of SH-β-CD: (**A**) appearance of SH-β-CD; (**B**) morphology of SH-β-CD under the cryo-EM (×100); (**C**) particle size of SH-β-CD suspension at refrigeration at 4 °C; (**D**) zeta potential of SH-β-CD suspension under refrigeration at 4 °C; (**E**) particle size distribution map. The data are means ± SD (*n* = 3).

**Figure 3 ijms-24-15177-f003:**
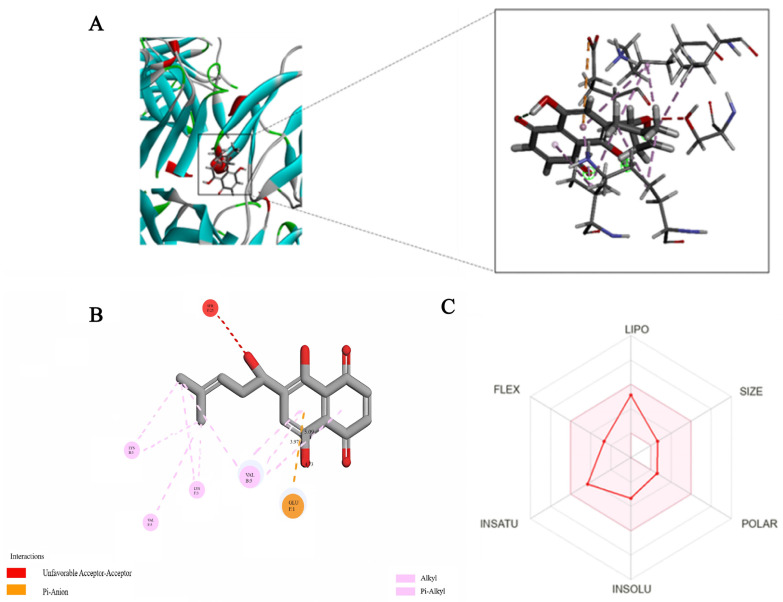
Molecular docking analysis: (**A**) on the left is the molecular docking diagram of MMP_S_ and SH, and on the right is the 3D-Molecules Docking MMPS (ID: 1JZH); (**B**) 2D-Molecules Docking; (**C**) bioavailability radar of the ligands were evaluated using the SwissADME web tool.

**Figure 4 ijms-24-15177-f004:**
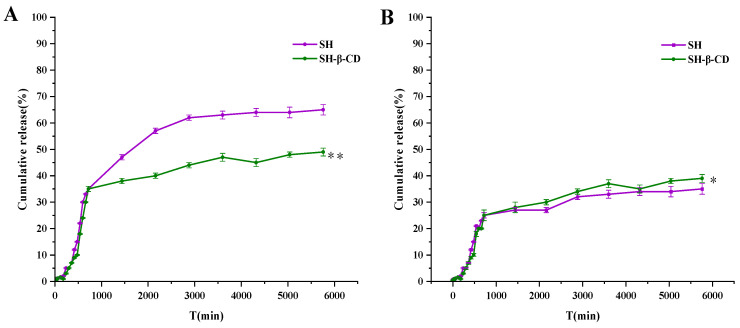
(**A**) Diffusion curves for SH and SH-β-CD in vitro. (**B**) Diffusion curves for SH and SH-β-CD ex vivo.(* *p* < 0.05, ** *p* < 0.01).

**Figure 5 ijms-24-15177-f005:**
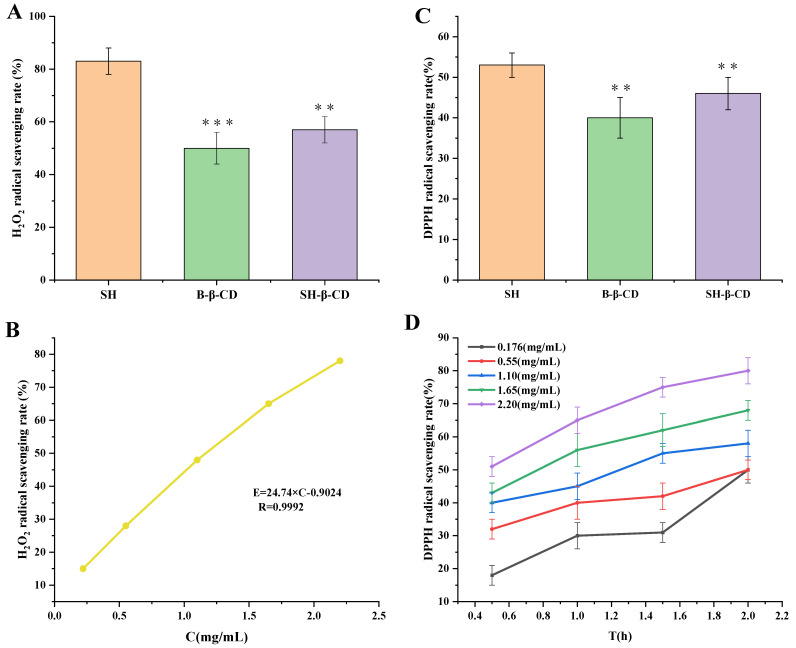
Free radical scavenging capacity. (**A**) Scavenging rate for 1.10 mg/mL SH, B-β-CD, and SH-β-CD on H_2_O_2_ free radicals. (**B**) Scavenging rate for different concentrations of SH on H_2_O_2_ free radicals. (**C**) Scavenging rate for SH, B-β-CD, and SH-β-CD to DPPH free radicals, respectively. (**D**) Scavenging rate for different concentrations of SH (0.176, 0.55, 1.10, 1.65, 2.20 mg/mL) on DPPH radicals. The data are means ± SD (*n* = 3). (** *p* < 0.01, *** *p* < 0.001).

**Figure 6 ijms-24-15177-f006:**
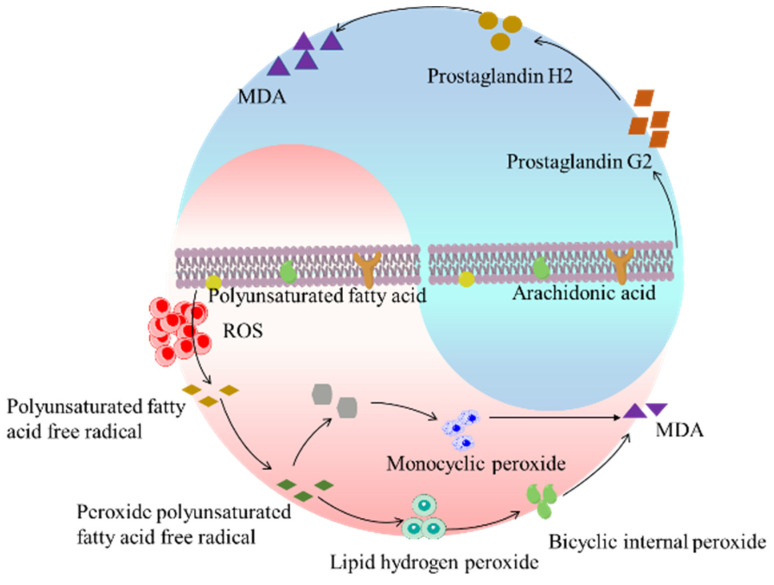
The pathway of MDA generation.

**Figure 7 ijms-24-15177-f007:**
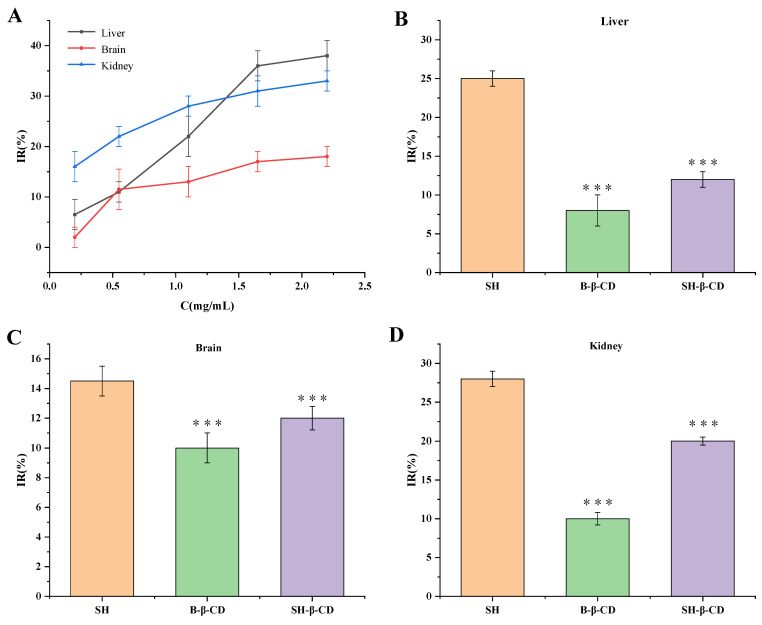
Inhibition of MDA production by SH, B-β-CD, and SH-β-CD in the liver, brain, and kidney homogenate of mice. (**A**) Inhibition rate of different concentrations of SH (0.22, 0.55, 1.10, 1.65, 2.20 mg/mL) on MDA in three homogenates. (**B**) Inhibition rate of SH, B-β-CD, and SH-β-CD on MDA in liver homogenate. (**C**) Inhibition rate of SH, B-β-CD, and SH-β-CD on MDA in brain homogenate. (**D**) Inhibition rate of SH, B-β-CD, and SH-β-CD on MDA in kidney homogenate. The data are means ± SD (*n* = 3) (*** *p* < 0.001).

**Figure 8 ijms-24-15177-f008:**
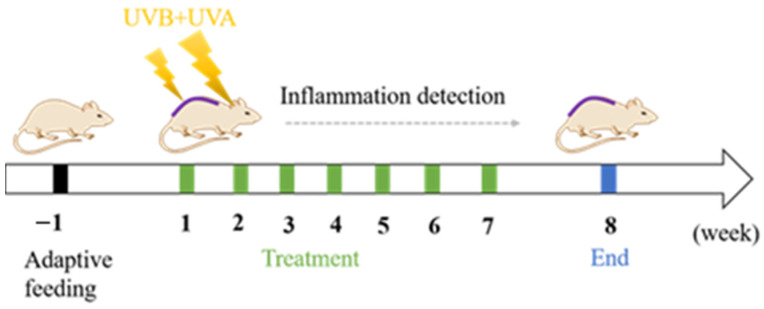
Antiphotoaging treatment scheme (*n* = 10).

**Figure 9 ijms-24-15177-f009:**
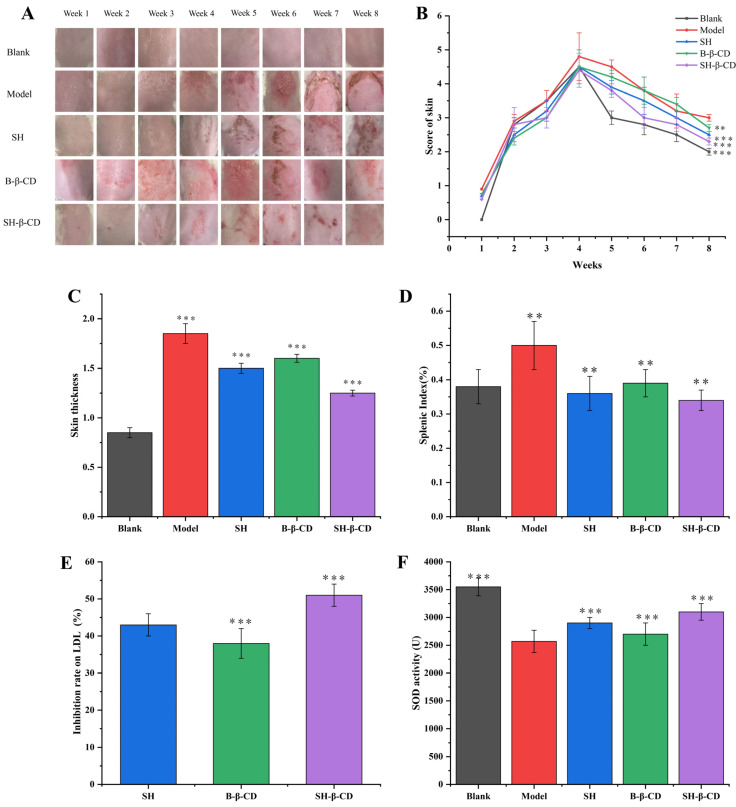
Skin condition of photoaged mice. (**A**) back skin injury of mice in each group; (**B**) score of skin; (**C**) skin thickness; (**D**) splenic index; (**E**) inhibition rate for SH and SH-β-CD on LDL in serum; (**F**) SOD activity in mouse skin homogenates. The data are means ± SD (*n* = 3) (** *p* < 0.01, *** *p* < 0.001).

**Figure 10 ijms-24-15177-f010:**
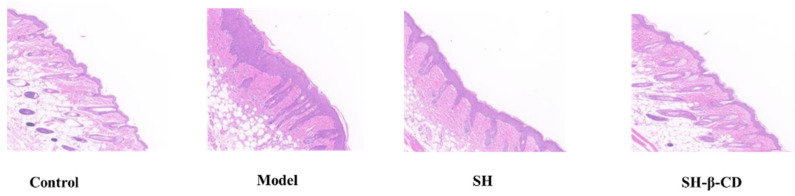
H&E staining to evaluate the histology of skin tissue. The tissue image is displayed under 100× magnification. Data are shown as mean ± SD.

**Table 1 ijms-24-15177-t001:** ADMET properties of SH and β-CD.

ADMET Predicted Profile—Classifications	SH	β-CD
Blood–Brain Barrier	+	+
Absorption Level	III	IV
Absorption Level	+	+
CYP2D6 inhibition	+	−
PPB Prediction	+	−
Skin Irritation	+	−

where “+” indicates that the effect is present, “−” indicates that the effect is absent, “III” indicates that it is low, and “IV” indicates that it is very low.

**Table 2 ijms-24-15177-t002:** Release and permeation parameters.

Category	Group	Zero Order	First Order	Higuchi	Weibull CDF	Hixson–Crowell
In vitro release parameters	SH	R = 0.8752	R = 0.5234	R = 0.9453	R = 0.9435	R = 0.9354
	SH-β-CD	R = 0.7767	R = 0.8226	R = 0.9021	R = 0.9824	R = 0.9262
Ex vivo permeation parameters	SH	R = 0.6034	R = 0.9117	R = 0.8258	R = 0.9768	R = 0.8626
	SH-β-CD	R = 0.5360	R = 0.9579	R = 0.7616	R = 0.9601	R = 0.8613

**Table 3 ijms-24-15177-t003:** Mathematical model of SH and SH-β-CD.

Model	Equation
Zero-order	Q = a + bt
First-order	Q = a × (1 − e − bt)
Higuchi	Q = at^1/2^ + b
Hixson–Crowell	Q = 100[1 − (1 − at)^3^]
Weibull CDF	Q = 1 − e^−(t/a)b^

where Q is the cumulative release rate; a and bare the drug release rate constants; and e is the exponential function, which ranges from 0 to 1.

**Table 4 ijms-24-15177-t004:** Grouping design for drug administration in animal experiments.

Group (*n* = 10)	Administration
Control	PBS (pH 7.4)
Model	PBS (pH 7.4)
SH	1.10 mg/mL of SH
β-CD	1.10 mg/mL of β-CD
SH-β-CD	1.10 mg/mL of SH-β-CD

**Table 5 ijms-24-15177-t005:** Macroscopic evaluation criteria for skin photoaging.

Score	Characterization
0	No wrinkles or sagging; longitudinal fine normal texture visible along the head and tail of the mice
1	Small wrinkles visible
2	Difficult to see the normal texture of the skin, full of tiny wrinkles
3	More shallow wrinkles are visible
4	Few deep wrinkles and mild sagging
5	Deep wrinkles increase significantly
6	Severe wrinkles; skin damage conditions

## Data Availability

Data are contained within the article.

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
