# Peer review of "Study on the Antioxidant Effect of Shikonin-Loaded β-Cyclodextrin Forming Host–Guest Complexes That Prevent Skin from Photoaging"

_ijms, 2023, doi:10.3390/ijms242015177_

Round 1

Reviewer 1 Report

Review of paper entitled: Study on the antioxidant effect of shikonin-loaded cyclodextrin

International Journal of Molecular Science

General Comments

The paper entitled “Study on the antioxidant effect of shikonin-loaded β-cyclodextrin forming host-guest complexes on preventing skin from the photoaging” describes the influence of the naphthoquinone active from Lithospermum erythrorhizon, called shikonin, in various in vitro, ex vivo and in vivo models focused principally on antioxidant effects. The paper suffers from numerous grammatical errors and paragraph structural problems that make reading the paper difficult.  If the authors wish to gain publication of the paper, they will need to have it reviewed closely by someone who can help to improve the grammar and syntax of the paper.

Unfortunately, the paper suffers from one significant problem that permeates throughout the numerous studies the authors undertook to demonstrate that encapsulating the skikonin in a β-cyclodextrin improves the performance of the bioactive molecule, they neglect to include the β-cyclodextrin as a control in their studies.  By overlooking the possible impact of the β-cyclodextrin alone in most of their studies, they assume that the β-cyclodextrin is inert.  This is a significant mistake and one that cannot be overlooked on the quality of the science being examined. 

In addition, the images in the paper are very small making seeing important details very difficult to see.  The authors must expand the size of the Figures significantly. 

Because of some significant problems within the body of the paper, this paper is not suitable for publication without significant improvements.  For this reason, the referee will only comment on the body of the paper and not on the Material & Methods section, Conclusions or References.

Specific Comments

Abstract

The Abstract has numerous grammatical errors (too many for the referee to summarize)

Introduction

·         The introduction also contains numerous grammatical errors.

·         The sentence in Lines 38-39 is not correct.  Sunscreens have been shown to significantly impact the effects of sunlight.

Results and Discussion

Encapsulation efficiency (EE)

·         Figure 2B does not demonstrate that the shikonin is encapsulated in the β-cyclodextrin as suggested in the paragraph above {Lines 83-84]

·         The image is missing label “E”.

Encapsulation docking

·         The authors suggest that the shikonin molecule can bind to a portion of the MMPs.  This does not prove that the molecule will inhibit the activity of the protein.  That would need to be proven with kinetic assays.  Also, the images of the docking do not account for the impact of the β-cyclodextrin on the binding to the MMP protein.

ADMET prediction

·         This referee is unfamiliar with the ADMET tool and would assume many readers are unfamiliar with the tool which is suggested to be supported by SwissADMET.  Presumably, the web tool may allow for a cursory examination of a particular molecule’s safety profile.  However, the typical cosmetic safety studies done on new molecular entities are Epiocular, Epiderm and a Human Repeat Insult Patch Test (HRIPT). The authors are making considerable assumptions about product safety using an in silico modeling system.  In addition, the various symbols used in Table 1 (+. -. III and IV) are undefined and so effectively meaningless. 

In vitro and ex vivo penetration of SH and SH-β-CD

Penetration of SH and SH-β-CD across the dialysis membrane in vitro

·         The skin is not a dialysis membrane.  While the size of a particular molecule may partially influence how it permeates the skin’s lipid barrier, the skin is a lipid barrier.  For this reason, Franz Cell studies done using a dialysis membrane are effectively meaningless to determine if a molecule or a complex is permeating the skin.  A superior model for in vitro skin permeation is the Skin Parallel Artificial Membrane Permeability Assay [https://pubmed.ncbi.nlm.nih.gov/22326705/].

·         The two WeibullCDF images C and D in Figure 4 are effectively redundant to images A and B above. 

Penetration of SH and SH-β-CD across the mouse skin ex vivo

·         The discussion in Lines 160-162 do not support the author’s suggestions that encapsulating the shikonin within the hydrophilic β-cyclodextrin will somehow improve the permeation of the lipophilic shikonin molecule through the mouse skin lipid bilayer.  The authors suggest in Lines 163-167 that somehow, the encapsulated shikonin has a superior skin permeation rate verses the unencapsulated molecule alone.  Not only is the shikonin molecule much smaller in size to the β-cyclodextrin encapsulated molecule, but the β-cyclodextrin encapsulation turns the lipophilic molecule into a hydrophilic molecule which would make it even less prone to skin permeation through the skin’s lipid bilayer.

·         The image in Figure 5 does not demonstrate the skin’s lipid bilayer, it shows a phospholipid bilayer of a living cell.  It is not representative of the skin permeation being discussed.

Antioxidant assay

·         Lines 183-186 are very confusing and seem almost to be pseudoscientific jibberish

Scavenging rate of SH and SH-β-CD on OH free radicals of H2O2

·         All the antioxidant studies reported have a significant error by not providing the β-cyclodextrin as a control.  The conclusions in Figures 6 and 8 are flawed by not having the β-cyclodextrin control to ensure that the results are not coming at least partially from the cyclodextrin alone.

Pharmacodynamic studies of skin photoaging

·         The studies reported in this section of the paper are the most convincing that the encapsulation of the shikonin by the cyclodextrin improves the performance of the molecule in resisting photoaging effects on skin.  The graphical data shows statistical significance which is critical.  However, as noted earlier, the lack of a cyclodextrin control diminishes the quality of the scientific work and the conclusions being made.

·         Lines 267-268: Figure 9 does not show the “macroscopic improvement effect” of the SH and SH-β-CD, it shows the schematic of the skin testing protocol.

·         In Table 3 (which is not a table, it is a figure) the data lacks the cyclodextrin control so, again, is meaningless.  Without this critical control, the reader must assume that the benefits are coming from the encapsulation, and this is a serious error in the paper.

·         The data provided in Figure 10 is the most interesting data in the paper as it comes from the in vivo mouse studies.  It is not immediately clear in the description or figure legend how many mice were employed.  The reader must locate this important detail in the Method section. This should appear in Figure 9 and be clearly noted [N=10].

The language and grammar in the paper must be improved.

Author Response

Response to reviewer 1

Dear reviewer:

     We sincerely thank you for giving us an opportunity to revise the manuscript. We have studied comments carefully and have made correction which we hope meet with approval. Revised portions are marked with red in the paper. The main corrections in the paper and the responses to the reviewer’s comments are as follows:

General Comments

The paper entitled “Study on the antioxidant effect of shikonin-loaded β-cyclodextrin forming host-guest complexes on preventing skin from the photoaging” describes the influence of the naphthoquinone active from Lithospermum erythrorhizon, called shikonin, in various in vitro, ex vivo and in vivo models focused principally on antioxidant effects. The paper suffers from numerous grammatical errors and paragraph structural problems that make reading the paper difficult.  If the authors wish to gain publication of the paper, they will need to have it reviewed closely by someone who can help to improve the grammar and syntax of the paper.

Unfortunately, the paper suffers from one significant problem that permeates throughout the numerous studies the authors undertook to demonstrate that encapsulating the skikonin in a β-cyclodextrin improves the performance of the bioactive molecule, they neglect to include the β-cyclodextrin as a control in their studies.  By overlooking the possible impact of the β-cyclodextrin alone in most of their studies, they assume that the β-cyclodextrin is inert.  This is a significant mistake and one that cannot be overlooked on the quality of the science being examined. 

In addition, the images in the paper are very small making seeing important details very difficult to see.  The authors must expand the size of the Figures significantly. 

Because of some significant problems within the body of the paper, this paper is not suitable for publication without significant improvements.  For this reason, the referee will only comment on the body of the paper and not on the Material & Methods section, Conclusions or References.

Specific Comments

Abstract

The Abstract has numerous grammatical errors (too many for the referee to summarize)

Introduction

  • The introduction also contains numerous grammatical errors.

Thanks for the reviewer’s constructive suggestion.We have revised them carefully.

1.The sentence in Lines 38-39 is not correct.  Sunscreens have been shown to significantly impact the effects of sunlight.

Thanks for the reviewer’s constructive suggestion. In the article, we had changed the sentence. (lines 38-39)

2.Figure 2B does not demonstrate that the shikonin is encapsulated in the β-cyclodextrin as suggested in the paragraph above (Lines 83-84)

Thanks for the reviewer’s constructive suggestion. In the article, the function of Figure 2B appearing in lines 83-84 is that the inclusion compounds in the figure have a certain regular shape and spherical inclusion compounds without obvious adhesion around them can be observed by cryo-electron microscopy. The phase identification methods of inclusion compounds include: microscopy, scanning electron microscopy, solubility method, UV-visible spectrophotometry. In addition, in the encapsulation rate experiment, we measured the encapsulation rate of 65.9 ± 7.13% by UV spectrophotometry, further indicating that shikonin is encapsulated in β-cyclodextrin. The dialysis experiment also showed that SH was encapsulated in β-cyclodextrin and slowly released.

(lines 117-118)

3.The image is missing label “E”.

Thanks for the reviewer’s constructive suggestion. After careful research, we modified the image labels in the article so that they correspond to each other in context. (lines 117-118)

4.The authors suggest that the shikonin molecule can bind to a portion of the MMPs.  This does not prove that the molecule will inhibit the activity of the protein.  That would need to be proven with kinetic assays.  Also, the images of the docking do not account for the impact of the β-cyclodextrin on the binding to the MMP protein.

Thanks for the reviewer’s constructive suggestion. Through literature review, we found that SH can inhibits the expression and proteolytic activity of MMP-9. MMP-9 and MMP-2 are important ECM degrading enzymes. They are reported to be involved in the invasion and metastasis of cancer cells. We also found that SH inhibited the expression and promoter activity of MMP-9 in MDA-7 breast cancer cells. MMP-9 expression and promoter activity in MDA-MB-231 cells with high metastasis potential. These results suggest that SH inhibits the activity of MMPs. Since β-cyclodextrin is a macromolecule and no suitable site was found when binding to MMP protein, we focused on the binding of SH to MMPs. Source DOI: 10.3892/or.2014.3159

5.This referee is unfamiliar with the ADMET tool and would assume many readers are unfamiliar with the tool which is suggested to be supported by SwissADMET.  Presumably, the web tool may allow for a cursory examination of a particular molecule’s safety profile.  However, the typical cosmetic safety studies done on new molecular entities are Epiocular, Epiderm and a Human Repeat Insult Patch Test (HRIPT). The authors are making considerable assumptions about product safety using an in silico modeling system.  In addition, the various symbols used in Table 1 (+. -. III and IV) are undefined and so effectively meaningless.

Thanks for the reviewer’s constructive suggestion. After careful examination, we modified the results of ADMET. The evaluation methods in the table are explained. Where " +" represents this effect, "-" represents no such effect, "III" represents a low degree, and "IV" represents a very low degree. (lines 141-142)

6.The skin is not a dialysis membrane.  While the size of a particular molecule may partially influence how it permeates the skin’s lipid barrier, the skin is a lipid barrier.  For this reason, Franz Cell studies done using a dialysis membrane are effectively meaningless to determine if a molecule or a complex is permeating the skin.  A superior model for in vitro skin permeation is the Skin Parallel Artificial Membrane Permeability Assay [https://pubmed.ncbi.nlm.nih.gov/22326705/].

Thanks for the reviewer’s constructive suggestion. In the article, when we use Franz Cell diffusion cell for penetration testing, we use internationally recognized detection methods.

Sources

[1] Siqueira C M et al 2013 H3N2 homeopathic influenza virus solution modifies cellular and biochemical aspects of MDCK and J774G8 cell lines Homeopathy: J. Faculty Homeopathy 102 31–40

[2] de Campos V E, Teixeira C A, da Veiga V F, Ricci E Jr and Holandino C 2010 L-tyrosine-loaded nanoparticles increase Figure 5. Permeation profile of NE with chitosan (blue) and NE without chitosan (red) labeled with 99 m Tc. Error bars indicate SD for the triplicates. 8 Nanotechnology 27 (2016) 015101 C S Cerqueira-Coutinho et al the antitumoral activity of direct electric current in a metastatic melanoma cell model Int. J. Nanomed. 5 961–71

[3] Bartosova L and Bajgar J 2012 Transdermal drug delivery in vitro using diffusion cells Curr. Med. Chem. 19 4671–7.

[4] Bronaugh R L and Stewart R F 1985 Methods for in vitro percutaneous absorption studies IV: the flow-through diffusion cell J. Pharm. Sci. 74 64–7

[5] Franz T J 1975 Percutaneous absorption. On the relevance of in vitro data J. Investig. Dermatol. 64 190–5

A Franz chamber is a vertically diffused glass device used to assess the release, retention, and penetration of drugs into the skin. Release, retention and penetration of drugs in the skin. The system consists of donor chamber and recipient chamber. Separated by a synthetic membrane or skin. Or separated by skin. The system simulates what happens when the agent is applied to the skin. Of course, as you mentioned, skin parallel artificial membrane penetration test is also a good model for skin penetration in vitro.

7.The two WeibullCDF images C and D in Figure 4 are effectively redundant to images A and B above.

Thanks for the reviewer’s constructive suggestion. We had removed the C and D in Figure 4 from the article based on your comments.( lines 173-174)

8.The discussion in Lines 160-162 do not support the author’s suggestions that encapsulating the shikonin within the hydrophilic β-cyclodextrin will somehow improve the permeation of the lipophilic shikonin molecule through the mouse skin lipid bilayer.  The authors suggest in Lines 163-167 that somehow, the encapsulated shikonin has a superior skin permeation rate verses the unencapsulated molecule alone.  Not only is the shikonin molecule much smaller in size to the β-cyclodextrin encapsulated molecule, but the β-cyclodextrin encapsulation turns the lipophilic molecule into a hydrophilic molecule which would make it even less prone to skin permeation through the skin’s lipid bilayer.

Thanks for the reviewer’s constructive suggestion. In the article, we had changed the sentence from "The complex physiological structure of the skin is the main barrier to drug pene-tration. SH-β-CD can penetrate across the skin and accumulate in the skin layers, thus ensuring that the drug can exert local effects on the skin" to " Encapsulation of SH in β-CD improves drug stability and enables efficient loading. This can be verified in transdermal experiments "(161-162)

Through careful examination, we found that the physiological structure of the skin is complex and is the main obstacle to drug penetration. SH-β-CD penetrates the skin and accumulates in the skin layer, ensuring that the drug works locally on the skin. Instead, SH encapsulation in β-CD can improve the stability of the drug and achieve high efficiency loading. This can be verified in transdermal experiments. Because cyclodextrin is a macromolecule, the SH-β-CD formed after SH encapsulation is also a macromolecule. Because the macromolecule is endocytosis through the cell membrane, the time required for transmembrane transport is shortened, and then SH is further released from the inclusion compound, resulting in the penetration effect of SH-β-CD due to SH(lines 167-171)

9.The image in Figure 5 does not demonstrate the skin’s lipid bilayer, it shows a phospholipid bilayer of a living cell.  It is not representative of the skin permeation being discussed.

Thanks for the reviewer’s constructive suggestion. We had removed the figure from the article based on your comments.

10.Lines 183-186 are very confusing and seem almost to be pseudoscientific jibberish

Thanks for the reviewer’s constructive suggestion. In the article, we had changed the sentence from "Antioxidants have scavenging free radical ability to protect cells from oxidative damage. Antioxidants were captured by the chain reaction to generate new free radicals with low potential energy and made the very active free radicals into stability, leading to a chain reaction of transmission interrupt, thus protecting cells and tissue from its damage " to "When the body is damaged, under pathological conditions, oxidative stress damage and other reactions occur, free radicals will be generated in the cell, which will further produce toxic effects on the cell. SH acts as a natural antioxidant with the ability to scavenge free radicals, protecting cells from oxidative damage."(lines 184-188)

11.All the antioxidant studies reported have a significant error by not providing the β-cyclodextrin as a control.  The conclusions in Figures 6 and 8 are flawed by not having the β-cyclodextrin control to ensure that the results are not coming at least partially from the cyclodextrin alone.

Pharmacodynamic studies of skin photoaging

Thanks for the reviewer’s constructive suggestion. Through careful examination, we completed the experimental results of the β-cyclodextrin group in the experiment.

12.The studies reported in this section of the paper are the most convincing that the encapsulation of the shikonin by the cyclodextrin improves the performance of the molecule in resisting photoaging effects on skin.  The graphical data shows statistical significance which is critical.  However, as noted earlier, the lack of a cyclodextrin control diminishes the quality of the scientific work and the conclusions being made.

Thanks for the reviewer’s constructive suggestion. When preparing SH-β-CD, we prepared blank β-cyclodextrin B-β-CD without drug. Since β-cyclodextrin also has certain antioxidant capacity, in order to control variables, we also set the B-β-CD group in the subsequent antioxidant experiment, DPPH and OH free radical experiment. The effect of β-cyclodextrin alone on the antioxidant capacity was eliminated. Since the research focus of this paper was the change of antioxidant capacity after SH was made into SH-β-CD, we focused on the SH group and the SH-β-CD group.

  1. Lines 267-268: Figure 9 does not show the “macroscopic improvement effect” of the SH and SH-β-CD, it shows the schematic of the skin testing protocol.

Thanks for the reviewer’s constructive suggestion. In Figure 9, we only show the initial schematic diagram of the experimental process, mainly to help readers understand the general implementation steps of the experiment, and do not involve the effects of drugs and preparations on the mouse photoaging model. The macro improvement effects are further shown in Figure 8. (line 293)

  1. In Table 3 (which is not a table, it is a figure) the data lacks the cyclodextrin control so, again, is meaningless. Without this critical control, the reader must assume that the benefits are coming from the encapsulation, and this is a serious error in the paper.

Thanks for the reviewer’s constructive suggestion. We set the blank group at dialysis, the raw material group of shikonin, the β-cyclodextrin group without added medicine, and the β-cyclodextrin group. In the subsequent experiments, the above groups were also set up respectively. Because the effect was not obvious, it was not reflected in the article. Instead, the emphasis was placed on the preparation, and it was discussed that the preparation group had a significant effect on the treatment of photoaging compared with the bulk drug. This has now been completed in Figure 9A. (line 353)

15.The data provided in Figure 10 is the most interesting data in the paper as it comes from the in vivo mouse studies.  It is not immediately clear in the description or figure legend how many mice were employed.  The reader must locate this important detail in the Method section. This should appear in Figure 9 and be clearly noted [N=10].

Thanks for the reviewer’s constructive suggestion. We have indicated the number of mice in Figure 8.(line 289)

Thanks!

Reviewer 2 Report

This is an interesting approach dealing with original antioxidant drugs as novel choice for topical treatment of skin photoaging. The manuscript is well designed and data and conclusions are sustained by thoroughly statistics analysis.

Some specific comments:

The term “in vitro” becomes a bit confusing because in cell biology tests the "in vitro" concept refers to testing compounds of interest on standard cell lines; please briefly explain this aspect. On the other hand, "in vitro" testing assumes the sterility of the compounds to be applied in the cellular system; please elaborate also a little on this aspect.

The mouse model comprises only females; why weren't male mice included? Please explain briefly this aspect. Additionally, it would be helpful for the manuscript accuracy if the authors could elaborate a bit on the age equivalent of mice relative to human age.

Please highlight what SH-β-CD brings new to the treatment of skin photoaging compared to current (approved) photoaging agents. Thus, the “Conclusion” section could be extended in relation to the novelty degree of SH-β-CD for therapeutic purposes in photoaging.

Author Response

Response to reviewer 2

Dear  reviewer:

     We sincerely thank you for giving us an opportunity to revise the manuscript. We have studied comments carefully and have made correction which we hope meet with approval. Revised portions are marked with red in the paper. The main corrections in the paper and the responses to the reviewer’s comments are  as follows:

  1. The term “in vitro” becomes a bit confusing because in cell biology tests the "in vitro" concept refers to testing compounds of interest on standard cell lines; please briefly explain this aspect. On the other hand, "in vitro" testing assumes the sterility of the compounds to be applied in the cellular system; please elaborate also a little on this aspect.

Thanks for the reviewer’s constructive suggestion. In the article , in vitro refers to outside the organism, and in addition, the series of operations we performed were carried out under sterile conditions.”.

2.The mouse model comprises only females; why weren't male mice included? Please explain briefly this aspect. Additionally, it would be helpful for the manuscript accuracy if the authors could elaborate a bit on the age equivalent of mice relative to human age.

Thanks for the reviewer’s constructive suggestion. After careful examination, Since female mice are relatively milder than male mice, we need to monitor the mice for a long period of time in the photoaging mouse model, and considering the maneuverability, choosing female mice is more conducive to the experiment. We know from the literature that 5 weeks to 17 months of age in mice is equivalent to 13 to 50 years of age in humans.

3.Please highlight what SH-β-CD brings new to the treatment of skin photoaging compared to current (approved) photoaging agents. Thus, the “Conclusion” section could be extended in relation to the novelty degree of SH-β-CD for therapeutic purposes in photoaging.

Thanks for the reviewer’s constructive suggestion. The approved photoaging agents are UV944 UV2908, SH-β-CD prepared in the present study has few side effects and at the same time it is easy to process the comfrey into other solid dosage forms such as tablets, capsules, dispersions, etc. after making the encapsulated compound, thus increasing the stability of the drug and improving the therapeutic effect.

Thanks!

Reviewer 3 Report

The manuscript by Yue et al. on shikonin/β-cyclodextrin is interesting despite the lack of dramatically new information. Regrettably, this version is unsuitable for publication due to the many inaccurate expressions, including the unresolved or resolved in the wrong place abbreviations and some missing or incomplete information. A thorough revision process may bring it into a publishable form.

At least the following concerns of the reviewer require attention:

- The authors failed to calculate the apparent complex stability constant. The calculation of this parameter could have been successful, as they practically recorded the solubility isotherm. The complex stability constant can also provide additional information about the guest release.

- Although the authors carefully carried out the experiments, they did not pay the same attention to writing the manuscript. The corresponding author failed to standardize the writing style.

- Some of the figures are very challenging. The "best" is in Figure 6b, where the authors have traced light yellow dots on a white background. On the other hand, although this figure is theoretically about a linear calibration curve, the authors did not draw the fitted line.

- In the experimental part, the authors often used the imperative mode, i.e., these sentences are commands to technicians, not descriptions of the methods used. Please check the whole section and transform the incorrect sentences into descriptive ones. Additionally, many sentences of that chapter are troubled (e.g., line 396).

- According to the Instruction for Authors, the journal names in the Reference section should be abbreviated, but the authors did not care about this requirement.

- In the reviewer's opinion, Figure 1 is better suited to a graphical abstract than a figure in the text.

- As Figure 12 contains nothing new, its removal is strongly recommended. The referee cannot understand its function and adds unnecessary bulk to the manuscript.

The following figures need improvements:

 - In Figure 1c, the molecular structure is not a chemical drawing software molecule. The violet color of chemical bonds is weird and useless. It would be better to stick to scientific standards. The figure quality is also low.

 - In Figure 1d, the left-side CD structure is strange and inappropriate, and the quality of the right-side CD structure is horrible. Additionally, these two pictures are meaningless.

 - Figure 2a must be a color picture because the presence of the current version is questionable.

 - The information in Figure 3b is invisible, as the texts are unreadable. The 2D-Molecular docking expression is hard to interpret, and the current use is meaningless. The 2D in terms of a protein does not exist, and the 2D representation of a molecule is unusable for any docking simulation. Probably, the authors would like to express a 2D drawing of the docked state.

- Figure 3c requires some explanation. What is the meaning of INSATU and INSOLU?

 - The thin green line in Figure 4b is almost invisible. As the authors could use thick lines in Figure 4a, they could have done the same in Figure 4b. On the other hand, the Figures 4c and 4d legends do not agree with the contents (they are lines only, without data points).

 - The removal of Figure 5 is mentioned above.

 - The light yellow data points and the missing line have been mentioned above.

 - The function of colored regions in Figure 7 is unknown.

 - The legend of Figure 8 mentions A/B/C/D subfigures, but the presented version does not contain identification letters.

 - The unreadable light yellow text in Figure 9 is funny. By the way, the UV light is more violet than yellow.

 - In the legend of Figure 10, the authors wrote weight change for Figure 10c. As far as the referee knows, the mass unit is not a centimeter or millimeter. Additionally, the size of an organ does not always correlate with its weight.

 - The unnecessity of Figure 12 has been mentioned above.

- The abbreviations should be defined upon their first appearance:

ROS (line 28); EM (line 81 or 82); MMPs redefinition (line 119); (weibull)CDF (line 174); DPPH (line 187); TTAGGG (line 236);  H&E (line 352 or 353); usually, C0 means the starting (weighed-in) concentration and not the found value; although ADMET is a common abbreviation, supposedly needs definition; meaning of ?/?????? is unclear (below line 604)

- The lines 489-490 and 497-498 are identical.

- Lines 46 and 49 seem incorrect.

- The αCD has a smaller cavity than βCD, despite the author's statement in lines 55-56 ("Compared with α-CD and γ-CD, β-CD has a moderate size of cavity ..."), but both α- and γCD are suitable for complexation. The authors' scheme suggests that they assumed the alkylene chain approaches the cavity from the primary hydroxyl side of βCD. Usually, αCD prefers the alkyls, and γCD the bicyclic moieties. Why did the authors assume that βCD is the most suitable host molecule? Do they have information from previous studies?

- Line 58, according to the authors, βCD molecules trap the poor water-soluble guest molecule. The authors are probably unaware that βCD can trap water-soluble molecules (water itself, organic and inorganic salts, water-soluble alcohols, etc.).

- Line 61, CDs do not increase the solubility of the guest! The complex has a higher solubility than the naked guest molecule, so its concentration is higher only, but not its solubility.

- Various spelling of cryo-EM.

- The cavity of CDs is not hydrophobic, only significantly less hydrophilic than the hydroxyl rims. There are many publications about the attempts to measure/determine cavity hydrophilicity and polarity.

- What did the authors express in lines 86-87? Does the self-assembly mean the aggregation of complexes? If so, use aggregation instead of self-assembly because the meaning is different.

- Two decimal digits in the hundred scale of dimensions is incorrect. Additionally, as the particle size calculation from the scattered light has a relatively high experimental error, even the one decimal digit can be exaggerated or meaningless.

- Line 90 contains an unknown unit (Mv). The prefix M stands for mega, and the referee has doubts that zeta potential is in the mega range, whatever the lowercase v stands for.

Both the host and guest are neutral molecules. What does "similar charges" mean? Additionally, the zeta potential is near the limit of the unstable colloid state (the "good stability" starts over |25-30| mV), and the uniform distribution has minimal connection to the zeta potential. The size distribution polydispersity has more significant information about particle size uniformity than zeta potential.

- The authors failed to mark the particle size distribution graph with "E" (the "E" is mentioned in the caption). It is unclear whether the presented particle size distribution is the lognormal distribution or the envelope curve of the multimodal distribution. As the lognormal distribution is a calculated curve, the multimodal version can provide information on the frequency of the particle sizes calculated, which are rarely symmetric.

- In lines 121-123, the spelling of amino acids is inconsistent. In the three-letter version, names start with a capital letter, and the rest is lowercase.

- At 175, the simulation curve is more of a fitted curve. On the other hand, the curve fitting is not a simulation.

- In line 176, the WeibullCDF model has not higher but the highest R-value in most of the cases. In the first row of the table, the minimal differences in R-values between the Higuch and WeibullCDF models are insignificant.

- In line 193, the meaning of the "O" band lone pair electron is unclear. What does the "O" band mean?

- The preparation of a suspension from βCD is hard to interpret. Furthermore, 1.2 g βCD/10 mL + 1 mL SH/EtOH does not give a complex solution of 1.1 mg/ml. The authors only mixed the components at an unknown concentration of SH. The authors also failed to insert the amount of EtOH used, so the 1 mL SH solution has no meaning.

The use of abs. EtOH is not understandable. Abs. EtOH always contains traces of benzene - unless the authors have used an uncommon anhydration method. Nobody does this except organic preparative chemists, and they rarely do it. Benzene is a UV-active material, a good guest for the βCD, and toxic for cells.

The rpm/min is an acceleration and not a rotation unit.

- Check and compare the spelling of cryo-EM and zeta potential with the previously used versions.

- The r/min is an unknown unit. Please use the conventional rotational unit.

- The molecular docking procedure is incomplete and cannot be reproduced.

- Line 439 needs some revision.

- In Table 4, the first-order kinetic equation is not correct. The meaning of "e" in that linear equation is unclear.

- Line 515, again, the unit is incorrect.

- In Table 5, 1.10 mg/mL SH solution is written, but solvent is missing. In PBS, it is impossible to reach that concentration (SH is soluble near 0.16 mg/mL in a 1:5 solution of DMF-PBS, but that solution is unstable).

- In the conclusion chapter, the authors wrote that they had prepared the SH/βCD complex by recrystallization, which is factually incorrect. The authors used the precipitation/crystallization method in the complex preparation. Recrystallization means repeated crystallization to reach a higher purity or better enantiomeric ratio.

- The reformulation of the references is already mentioned. Please follow the journal recommendations. The "p." in journal citation is uncommon and useless.

The manuscript requires review by a native English chemist to ensure accuracy and technical writing quality.

Round 2

Reviewer 1 Report

The paper that has been revised after the referee's comments were addressed is a much better paper.  In particular the author's ability to include details on the Beta-Cyclodextrin has addressed one of my major concerns.  A few additional minor corrections and the paper should be suitable for acceptance for publication.

2. Results and discussion

Line 80;  "...host to guest of 1:1 and [improved solubility] in water..."

2.4 ADMET predictions

Table 1.  Absorption Leve[l] 

2.7.2. Scavenging rate....

Line 213: "...which [means] the SH cannot be completely...."

Figure 5-Please review the descriptions for Legend 5 as it seems that the labeling for (A) is still not quite correct.  Review this entire legend description carefully.

2.7.3. Inhibitory effect...

Line 239: "...harmful skin [problems] and even...."

Line 261: "...which [means] SH cannot..." 

2.8.  Pharmcodynamic studies...

The authors need a sentence describing Figure 8 that should appear at the beginning of the first paragraph before the refence to Reference 9A.

Author Response

Response to reviewer 1

Dear reviewer:

     We sincerely thank you for giving us an opportunity to revise the manuscript. We have studied comments carefully and have made correction which we hope meet with approval. Revised portions are marked with red in the paper. The main corrections in the paper and the responses to the reviewer’s comments are as follows:

  1. The paper that has been revised after the referee's comments were addressed is a much better paper. In particular the author's ability to include details on the Beta-Cyclodextrin has addressed one of my major concerns. A few additional minor corrections and the paper should be suitable for acceptance for publication.

Thanks for the reviewer’s constructive suggestion. After careful inspection, we have corrected the error as required.

  1. Results and discussion

Line 80; "...host to guest of 1:1 and [improved solubility] in water..."

Thanks for the reviewer’s constructive suggestion. The results show that the inclusion compound with 1:1 ratio of host and guest is formed and improved solubility in water. (line 80)

  1. 2.4 ADMET predictions

Table 1. Absorption Leve[l] 

Thanks for the reviewer’s constructive suggestion. After careful inspection, we have corrected the error as required.

  1. 2.7.2. Scavenging rate....

Line 213: "...which [means] the SH cannot be completely...."

Thanks for the reviewer’s constructive suggestion. After careful inspection, we have corrected the error as required.

5.Figure 5-Please review the descriptions for Legend 5 as it seems that the labeling for (A) is still not quite correct.  Review this entire legend description carefully.

Thanks for the reviewer’s constructive suggestion. After careful modification, in the article we have changed the label A in Figure 5 to the correct form.

  1. 2.7.3. Inhibitory effect...

Line 239: "...harmful skin [problems] and even...."

Line 261: "...which [means] SH cannot..." 

Thanks for the reviewer’s constructive suggestion. After careful inspection, we have corrected the error as required. (line 239,261)

  1. 2.8. Pharmcodynamic studies...

The authors need a sentence describing Figure 8 that should appear at the beginning of the first paragraph before the refence to Reference 9A.

Thanks for the reviewer’s constructive suggestion. After careful inspection, we have added the sentence describing Figure 8 in the article. 

Thanks a lot.

Reviewer 3 Report

The authors have corrected their manuscript correctly. 

Only one point remained unclear, but the manuscript does not require revision. Authors can correct it during the proofreading.

The authors did not understand the referee's message about the "unknown" unit, rpm/min. The authors answered the concerns correctly that rpm means rotation per minute. But, what they wrote in the manuscript was rpm/min=>rotation per minute per minute=>rotation per minute^2. The rotation/min^2 is an acceleration unit. This illegal unit is in lines 475, 547, and 604. In the newly inserted text in line 402, rpm is in upper case.

The simultaneous problem is the using r/min for rpm. The referee cannot understand why the authors cannot use units consistently.

The authors should fix these issues in the proofreading.

A minimal spell check is advised.

Author Response

Response to reviewer 3

Dear reviewer:

     We sincerely thank you for giving us an opportunity to revise the manuscript. We have studied comments carefully and have made correction which we hope meet with approval. Revised portions are marked with red in the paper. The main corrections in the paper and the responses to the reviewer’s comments are as follows:

  1. A few additional minor corrections and the paper should be suitable for acceptance for publication. The authors have corrected their manuscript correctly.

Only one point remained unclear, but the manuscript does not require revision. Authors can correct it during the proofreading.

The authors did not understand the referee's message about the "unknown" unit, rpm/min. The authors answered the concerns correctly that rpm means rotation per minute. But, what they wrote in the manuscript was rpm/min=>rotation per minute per minute=>rotation per minute^2. The rotation/min^2 is an acceleration unit. This illegal unit is in lines 475, 547, and 604. In the newly inserted text in line 402, rpm is in upper case.

Thanks for the reviewer’s constructive suggestion. After careful inspection, we have corrected the error as required. (lines 402,475,547,604)

2.The simultaneous problem is the using r/min for rpm. The referee cannot understand why the authors cannot use units consistently.

Thanks for the reviewer’s constructive suggestion. After careful inspection, we have unified the units.

3.The authors should fix these issues in the proofreading.

Thanks for the reviewer’s constructive suggestion. After careful inspection, we have corrected the error as required.

4.Comments on the Quality of English Language

A minimal spell check is advised.

Thanks for the reviewer’s constructive suggestion. After careful inspection, we have corrected the error as required. 

Thanks a lot.